# Experiencing one's own body and body image in living kidney donors–A sociological and psychological study

Katarzyna Kowal[1]*, Mateusz Zatorski[2], Artur Kwiatkowski[3]

**1** Chair of Health Science and Physiotherapy, Jan Dlugosz University in Czestochowa, Czestochowa, Poland, **2** Department of Clinical Psychology and Health, University of Social Sciences and Humanities, Poznan, Poland, **3** T. Orlowski Transplant Institute of Medical University of Warsaw, Poland, The County Hospital in Ketrzyn, Ketrzyn, Poland

☯ These authors contributed equally to this work.
* k.kowal@ujd.edu.pl

**Data Availability Statement:** All relevant data are within the paper and its Supporting Information files.

**Funding:** The authors received no specific funding for this work.

## Abstract

### Background

The aim of the study was to perform an in-depth exploratory analysis of the experience and image of one's body among living kidney donors.

### Method

The research was carried out using mixed methodology. The study on experiencing one's own body was carried out using the sociological methodology of the grounded theory (qualitative research). This method was supplemented with psychometric measurement–the Body Esteem Scale (quantitative research). The basic research method was the in-depth interview. Using this method, a group of 25 living kidney donors who had not experienced any serious health or psychological problems after donation was examined. The participants of the study came from three transplant centers in Poland.

### Results

The data from the sociological interviews indicate that the donors: 1. do not experience radical changes in the functioning of their body; 2. maintain full control over it and do not feel the absence of a kidney in the body; 3. consciously and reflectively take care of their body after donation. In addition, the sociological research indicates that caring for one's own body also includes the transferred organ. The kidney donors experience a kind of bodily identity extension, including the recipient's body. However, the personal and social identity of the studied kidney donors is not disturbed in any way. The psychometric data correspond to the sociological results and indicate: 1. a lack of extreme emotional assessments about one's body; 2. awareness of one's own body and consistency of its image; 3. reduced emotional assessment of body zones directly related to the surgery; 4. differences in body image between the sexes.

**Competing interests:** The authors have declared that no competing interests exist.

## Conclusions

The research results presented in the text indicate not only the possibility, but also the need for triangulation of research methods in the study of the experience and image of one's own body in living kidney donors. The proposed research approach employing mixed methodology within the fields of sociology and psychology for researching the phenomenon of living kidney donation is not very common.

## Introduction

The development of modern knowledge and medical technology means that in the 21st century the body is increasingly moving from the sphere of nature to that of culture. Thus, the human body is increasingly becoming an object of interest for representatives of the humanities and social sciences. The issue of kidney transplantation from a living donor seems to be a good example of this for at least two reasons.

First, kidney donation from a living donor is increasingly being treated as a standard medical procedure. The data from studies carried out by various transplant centers on large groups of donors indicate a low risk of complications during and after the donation procedure [1]. The research results also indicate a low risk of mental health disorders in living donors, which occur most frequently in the short postoperative recovery period [2–4]. The key aspect to be controlled is the quality of life of the donors. Research shows that the quality of life of donors should be viewed from two perspectives: a. physical well-being and b. mental well-being, as well as life satisfaction. Both physical and mental well-being are reduced in the period close to the donation procedure [5]. If the donors do not belong to a risk group, i.e. people experiencing complications after surgery, old age or who are overweight [6], they regain satisfaction with their health after about 12 months [7]. More inconclusive research results relate to the mental well-being of donors. Some studies indicate that donors may feel permanent regret after kidney donation if the transplant procedure did not bring the expected improvement in the recipient's health or the donor himself/herself cannot return to his/her social roles [8]. Nonetheless, other studies indicate that donors return to the baseline level of life satisfaction [9]. Some data indicate possible personal development after a crisis situation such as donation [10]. By registering the consequences of this type of medical procedure, increasingly more attention is paid to the individual perspective of the patient. Monitoring the donor's status goes beyond physical examination, laboratory results, or information from diagnostic imaging. Determining the level of their well-being forces one to reach for the methodology of the humanities.

Secondly, the kidney donation decision should be viewed from a broad perspective. This means taking into account not only the basic criterion, which is having sufficient health resources, but also the context of currently performed roles and social relations [11]. Factors that can also affect donation motivation include the religious systems and legislation in which the donor operates [12, 13]. The background of material support provided to donors is also important [9]. All these aspects affect the assessment of one's body after the donation procedure. Identifying the changes occurring in the body after kidney donation is therefore closely related to cultural influence in the broad sense. Hence, the current definitions of body experience and image include social, cultural and biological components [14].

The presented research project is part of the body research field which has been interpreted as part of the sociological and psychological reference system. As the substantive justification

for the conducted research, we point to the socio-cultural conditions of treatment of the body in the 21st century [15–20]. In the context of the donation procedure, the following deserve special attention: 1) post-modern lifestyle cultivating the body; 2) taking increasingly more control over the body; 3) extension of the scope of the right to decide about one's own body; 4) defining identity through the body.

The main research question that the project team members tried to answer was: How does interference with the human body, involving the removal of a kidney, affect the donors' experience and body image?

The aim of the presented research, located on the border of medicine, sociology and psychology, was:

1. to verify the effectiveness of the mixed methodology, using sociological and psychological tools, in examining bodily experiences related to kidney donation;

2. to identify the consequences of kidney donation for donor body experience and image.

A review of the literature indicates that studies of the body experience and image of living kidney donors are comparative. They relate to the consequences for the body image depending on different organ procurement techniques [21, 22], or the differences between the kidney recipients and their donors [23]. The vast majority of studies use a psychometric questionnaire based on measuring the overall body image and appearance changes after surgery [21, 22, 24, 25]. According to the authors of the project, noticing a broader perspective of the impact of medical treatments on the functioning of the patient is associated with the necessity to use the methodologies of various disciplines and to verify the research tools used within them. This was the first goal of the presented research. Other researchers have also pointed out the need to develop effective diagnostic tools corresponding to the complex context of experiencing donation [23].

The presented project is dominated by the phenomenological perspective of body research as a source of human knowledge and experience [26]. The phenomenological tradition, which the method of grounded theory draws from, is becoming increasingly common in journals, including those with a medical profile [27–31]. The main research method, which is the grounded theory method in the presented research, made it possible to go beyond previously used methods and techniques for understanding the consequences of donation. The choice of grounded theory methodology [32, 33] as a strategy for collecting and analyzing data was dictated primarily by the fact that it allows access to the category of experiencing one's own body in the context of donation. The motive for choosing grounded theory methodology was also the promise for the researcher to obtain rich qualitative data, that is, data that reveal the thoughts, feelings, intentions and actions of the subjects, as well as their contexts and structures of everyday life [33]. This picture was to be completed by psychometric data limited to the emotional attitude towards one's body and satisfaction with its functioning [34, 35].

The use of mixed research methods with a qualitative and a quantitative component also meets the second objective of the study. The adopted data collection and analysis procedure can be described as the *funnel approach*. Qualitative sociological research enabled the study participants to take on the role of explorers of knowledge about their own corporeality after donation. This is the essence of research based on adopting the patient's perspective [36], which has not been described in the literature in the context of kidney donation. The psychometric study provided a different kind of data, based on focusing the attention of the subjects on individual parts of their body. The donors were mobilized to determine the attractiveness and daily functionality of their body. The sociological and psychometric research strategies, due to their methodological integration, provided the researchers with complementary data

on the experience and body image of kidney donors. This is a response to the needs formulated in the literature on the subject. The first one concerns emphasizing the subjective nature of experiencing one's own body and the changes taking place within it [37]. The second one appeals to the significant influence of the donors' body image on their daily functioning [11, 38].

## Material and methods

### Study participants

The study was conducted in a group of living kidney donors (N = 25). The subjects remained under the care of three transplant centers: the Infant Jesus Clinical Hospital in Warsaw, the Independent Public Provincial Integrated Hospital in Szczecin, and Dr. Antoni Jurasz University Hospital No. 1 in Bydgoszcz (organ procurement procedure, short- and long-term care of the donor after nephrectomy). The study was conducted outside the indicated units.

The selection of the respondents was deliberate and was determined by the criterion of being a kidney donor, with a minimum period of one year after donation. The study participants were people from three groups representing the forms of donation available in the Polish legislation system: 1. genetically related donors; 2. not blood-related but emotionally related donors; 3. donors who are strangers to the recipient in the paired exchange system. The socio-demographic characteristics of the respondents are presented in Table 1.

A feature describing the donors, which is extremely important from the point of view of the topic of the work, is the method of kidney procurement. The study group included 19 donors who underwent donation by the laparoscopic approach (76%) and 6 donors who underwent donation by the open approach (24%).

The donation procedure is an operation under general endotracheal anesthesia which involves the removal of a kidney for transplantation into an organ recipient. This procedure violates the continuity of the tissues depending on the surgical method: the open approach vs the laparoscopic approach. The latter is the gold standard. The use of the open approach is not justified and may be regarded as malpractice by many transplantologists. The difference between the two methods is the location of the operation site: incision under the costal arch vs a Pfannenstiel incision or in the midline below the navel. Each of these methods also has a different intensity and duration of pain. However, the feeling of interference with the organism's homeostasis by surgery should decrease over time in both cases.

The location of the live donor kidney scar will vary depending on the method used. In the open approach, it will be a cut under the costal arch, which is associated with greater pain intensity and longer exposure of the patient to pain. A larger cut also means a longer scar. In the laparoscopic method, it will most often be a transverse incision over the symphysis pubis (Pfannenstiel). In this approach, the incision is smaller, the pain is less, and the scar is smaller. In the case of the laparoscopic approach, the fact that there is no incision and thus no scarring of the abdominal cavity is important. The transverse cut above the pubic symphysis generates a scar that effectively is covered by the lower part of the underwear. In the long run, the differences in relation to the two methods become blurred and eventually cease to be relevant.

Medically, nothing happens with the site after the kidney is removed. There is no "black hole" left. The retroperitoneal space in which the kidney is located is limited by the peritoneum (a thin and flexible membrane), thanks to which intraperitoneal organs, including those constantly moving the intestine, occupy it.

The functioning of the body as a whole, as well as of individual parts of the body, should not change after an uncomplicated kidney donation procedure from a living donor (after the

**Table 1. Sample socio-demographic profile.**

|  | N | Minimum | Maximum | *M* | *SD* |
|---|---|---|---|---|---|
| **Donor's age** | 25 | 27 | 75 | 45.72 | 11.524 |
| **Time since donation in months** | 25 | 22 | 50 | 40.12 | 8.378 |
|  | N | Percentage | | | |
| **Gender:** | | | | | |
| male | 9 | 36 | | | |
| female | 16 | 64 | | | |
| **Marital status** | | | | | |
| married / in a long-term relationship | 19 | 76 | | | |
| single / not in a long-term relationship | 6 | 24 | | | |
| **Education:** | | | | | |
| higher | 11 | 44 | | | |
| secondary | 11 | 44 | | | |
| vocational and primary | 3 | 12 | | | |
| **Employment status:** | | | | | |
| employed | 20 | 80 | | | |
| unemployed | 5 | 20 | | | |
| **Place of residence:** | | | | | |
| city > 100,000 inhabitants | 10 | 40 | | | |
| town with 50,000–100,000 inhabitants | 5 | 20 | | | |
| rural area | 10 | 40 | | | |
| **Relationship between donor and recipient** | | | | | |
| parent–child | 5 | 20 | | | |
| child–parent | 1 | 4 | | | |
| siblings | 3 | 12 | | | |
| spouses | 8 | 32 | | | |
| daughter-in-law–mother-in-law | 1 | 4 | | | |
| strangers in the paired exchange programme | 7 | 28 | | | |
| **Kidney procurement method** | | | | | |
| Open approach | 6 | 24 | | | |
| Laparoscopic approach | 19 | 76 | | | |

wound has healed and during the rehabilitation period). There is no substantive justification for this in terms of human anatomy and physiology.

## Examination procedure

The presented research was carried out using methods from the fields of sociology and psychology. They included qualitative interviews, the material of which was analyzed in accordance with the procedures of grounded theory methodology, as well as a psychometric questionnaire that provided quantitative data.

At the outset, each of the participants gave their informed consent to participate in the study. Due to the adopted form of the study, informed consent was given verbally (carrying out the research by phone). The consent procedure was performed in several steps. The first was to provide information on the subject of the study (including the topic of the project), the people carrying it out, the course of the study and the specifics of each stage: first the qualitative interview and second the quantitative questionnaire. In the next step, information was provided about voluntary participation in the study, as well as its purely scientific nature.

Then, the participants were informed about their anonymity, which means that the participants can only be identified by the persons conducting the research and only for the purpose of analyzing the collected data. The respondents were also informed about the possibility of withdrawing from participation in the study at any time during its course, without giving the reasons for such a decision. At the very end, information was provided on the legal obligation imposed on the researchers to protect the obtained information (recordings of interviews and questionnaires). The data obtained during the study are protected by the Personal Data Administrator of the universities with which the researchers are affiliated. This procedure was repeated twice and preceded both implementation of the qualitative study and the collection of psychometric data. On this basis, the respondents made a decision about their participation in the study. The research was carried out by people who were in no way involved in the donation and transplantation procedures, who were the first two authors of the manuscript (KK–sociologist, MZ–psychologist). This was to limit the impact of factors not directly related to the main research subject, but interfering with the obtained data, e.g. positive or negative attitude to medical staff, a strong need for social approval or the impact of the information provided on the adopted form of care after surgery.

The survey was carried out from December 13, 2017 to September 30, 2018. It should be noted, however, that the sociological and psychometric tests did not take place at the same time. The psychometric testing stage took place on average about 6 months after the end of the sociological testing. Maintaining the time interval between the qualitative and quantitative studies was deliberate. The researchers wanted to avoid data contamination. The qualitative measurement concerned the experience of one's own body, and the quantitative measurement concerned its image. The authors hoped that the six-month interval would erase the memory trace of the interview content. At the same time, it was not long enough to involve significant changes in the basic identification of one's body. This allowed the independence of the two types of data to be maintained, and at the same time did not interfere with achievement of the set research goals.

## Consent of Bioethics Committee

The research project which is the empirical basis of this article is part of a broader research project entitled "Psychosocial dimension of relationship between donor and recipient after kidney transplantation". Project manager: Artur Kwiatkowski Prof. MD-PhD. Project location: T. Orłowski Institute of Transplantology at the Medical University of Warsaw. The project was submitted to the Bioethics Committee of the Medical University of Warsaw, 28 November 2017. In the statement attached to the manuscript of December 12, 2017, AKBE/234/2017, the Bioethics Committee at the Medical University of Warsaw acknowledges the information on the research and raises no objections to it. All the respondents gave their informed consent to participate in the study. The form of obtained consent was oral, and the witnesses were Mateusz Zatorski (consent of the respondents participating in the sociological part of the research) and Katarzyna Kowal (consent of the respondents participating in the psychometric study).

## Qualitative methods

The reporting of qualitative research was carried out in accordance with the COREQ criteria checklist [39].

Understanding how changes in the body under the influence of donation affect the way the donor experiences his own corporeality would not be possible without conducting a sociological study based on a qualitative strategy. As part of it, the methodology of grounded theory in

constructivist terms according to Kathy Charmaz was used [33]. An in-depth interview was conducted with each of the respondents, which, due to its interpretative potential, gives an opportunity to thoroughly investigate the experiences of donors and obtain rich empirical material. In the adopted variant of grounded theory methodology, the in-depth interview is called the intensive interview and is defined as an open, but targeted conversation, which, despite its specific form and pace, remains emergent and flexible [33]. The study of the individual experience of one's own body in kidney donors, which was the central analytical category of the sociological part of the project, focused not on the donor's description of physical changes occurring in his body as a result of donation, but on the subjective activity of the subjects in interpreting these changes. The in-depth interview was conducted in such a way as not to focus on the cognitive context of donation, but to strive to grasp the subjective meanings of donation in the context of the donors' experience of their own corporeality. Thus, how the subjects perceive and interpret their own body after donation was subjected to empirical exploration in the following dimensions: 1. somatic (the physical consequences of nephrectomy being experienced by the donor); 2. adaptive (attitudes towards one's own body, donor's lifestyle, scope of subjective agency and activity); 3. interactive and social (experiencing one's body at the level of social interactions in the micro dimension–in the family, social circle, in contacts with medical professionals); 4. identity (changes occurring in the personal identity of donors as a result of donations).

The person conducting the interviews with the kidney donors was the first author of this text (KK), a sociologist by profession specializing in: 1) sociology of health, disease and medicine; 2) sociology of the body; 3) qualitative sociology.

There were no interactions between the researcher and respondents prior to the start of the study. The motives behind the research project presented by the interviewer were purely substantive.

The sociological research was conducted from December 13, 2017 to February 28, 2018. The interviews were carried out by telephone. This form of contact with the respondents was determined by several important substantive considerations. First, the pilot study, which involved conducting three interviews through personal meetings and three interviews by telephone conversation, showed that the respondents were more likely to open up to the researcher during conversations excluding personal contact. Feeling more anonymous, the surveyed donors penetrated deeper into their experiences during the telephone conversations. Secondly, the respondents were residents of various regions of Poland, and reaching each of them physically was associated with high costs of the project. Thirdly, the authors consciously wanted to avoid conducting interviews in a hospital where the subjects entering the role of patients (during planned follow-up visits) might behave in a more restrained and controlled manner. Fourthly and finally, in a conscious way, the authors wanted to avoid conducting interviews during a meeting with the donor in his living space, which would deprive the study of confidentiality. All the donors invited to the study (N = 25) responded by agreeing to participate in the study. None of the invited participants declined to participate in the interview, nor ended it prematurely.

The in-depth interview was based on an interview outline previously prepared by the researcher, which contained some very general and open-ended questions about the issue under study. The questions concerned feelings, thoughts, actions, intentions and meanings related to specific situations and experiences previously described by donors. Attached is the list of the researcher's information needs, which was the tool to conduct the in-depth interview with the kidney donors (S1 Appendix).

The donors' statements were recorded using a voice recorder, to which each respondent agreed. During the interview, the respondents were most often in their own homes. The in-

depth interview lasted from 60 to 90 minutes. As a result of conducting 25 in-depth interviews, recorded audio material of 30 hours 56 minutes was obtained, which was then subjected to naturalized transcription–a copy of the spoken discourse.

Analysis of the collected empirical material was carried out in accordance with the principles of the grounded theory methodology of Kathy Charmaz [33], which determined the following stages of conceptual work: 1. verb coding using the "line by line" method; 2. concentrated coding to select the most analytically significant codes; 3. ascent to the level of conceptual categories; 4. construction of a theory grounded in data. One coder participated in the coding of the data, which was the first author of this work [KK]. According to the logic of qualitative research, the researcher's activity was not linear, and she was determined by her constant retreat to the empirical world until the moment of saturation of the properties of the theoretical categories. The procedures that proved to be the most useful in this regard were the constant comparative method and theoretical sampling.

## Quantitative methods

The choice of the employed tool was conscious and was based on a procedure consisting of several steps. The first of them analyzes current psychological theories devoted to the role of one's own body in the process of self-identification. There are currently three paradigms in psychology describing the experience of one's own body. The first of these is referred to as body image and is mainly associated with exteroception processes. The second is body schema, which in turn is based on information from proprioceptive sources [40]. The last, relatively new theoretical proposition concerns the sense of body states and is based on a wide range of interoceptive experiences [41]. Each of the paradigms refers to different levels of experiencing one's own body and also sets out a different methodology for studying these experiences. Body image refers to the level of the most conscious cognitive representations of one's own body and as such can be self-describing. This paradigm corresponds to the greatest extent with the approach represented in the sociological part of the research–the subjective, conscious perspective of the respondent. The second step in the tool selection procedure was based on a review of the psychometric tools used in body image studies, with particular emphasis on persons undergoing medical procedures. Most of the tools described in the subject literature are used to measure body awareness [42, 43], the socio-cultural factors responsible for body appearance [44, 45] or diagnosis of the consequences of clinical disorders [46]. The tools used in living donor research relate primarily to awareness and contentment with one's body. The BIS questionnaire is used in these studies [47]. The authors of the project wanted to identify both the conscious assessment of satisfaction with one's own body, not only in appearance, but also in its functioning. Another important criterion in selecting the tool was the possibility of distinguishing between the assessment of women and men. The literature on the subject also indicates differences between the sexes in experiences of their own body [48, 49]. The last step of the procedure was related to the search for a tool that would already have a reliable adaptation in the country where the research was carried out.

As a consequence of the adopted procedure, body image was studied using the Body Esteem Scale questionnaire [34]. Data were collected using the Polish version of the test in the adaptation by Małgorzata and Mariusz Lipowski [35]. The questionnaire consists of 35 questions. Each question concerns either a specific element of one's own body (e.g. *hips*, *lips*, *appearance of the eyes*) or body functions (e.g. *reflexes*, *appetite*, *sexual activity*). Each of these aspects is assessed in terms of the strength of positive versus negative emotions. A 5-point scale is used for assessment (from 1 –*I have strong negative feelings*, through 3 –*I have no feelings*, up to 5 –*I have strong positive feelings*).

The questionnaire consists of four subscales, two characterizing the body image separately for men and women. They are: for women–Sexual attractiveness (e.g. *lips*, *sex drive*), Weight Control (e.g. *figure*, *appetite*); for men–Physical Attractiveness (e.g. *chin*, *buttocks*), Body Strength (e.g. *width of shoulders*, *muscle strength*). For both sexes there is a fifth subscale–Physical Condition (e.g. *physical stamina*, *agility*). The results of the basic analyses of the psychometric parameters (reliability) were satisfactory and did not differ from the parameters obtained in the adaptation process; Cronbach's alpha 0.82 (for women) to 0.85 (for men).

The data collected as part of the psychometric part of the project were obtained via telephone with the donors or material sent electronically (so-called hard data). Individuals who, for reasons of time, preferred to send data via e-mail were also subject to the informed consent procedure. This was carried out in a short telephone conversation, during which the researcher received information about the e-mail address for correspondence. The psychometric tests lasted from March 1 to September 30, 2018. The person collecting the data for the psychometric part was the second author of this text (MZ), a psychologist by profession specializing in clinical psychology.

# Results

## Results of sociological research

Analysis of the empirical material accumulated in the sociological part of the research led to the emergence of the following analytical categories which make up the living body experience for living kidney donors: 1) experience of pain associated with kidney donation; 2) changes in body appearance after donation; 3) changes in body functions after donation; 4) the donor's experience of the lack of a kidney in the body; 5) the right of ownership of the kidney given to the recipient; 6) experiencing the kidney in the recipient's body; 7) self-definitions and body perception in kidney donors; 8) changes in the relation of the kidney donor to his or her own body after donation.

## 1. Experience of pain associated with kidney donation

Pain should be considered a common fact of experiencing one's own corporeality among all the studied kidney donors. What matters is the way the donors interpret the occurrence of pain. Despite admitting to experiencing strong pain in the first days after the surgery, the donors distance themselves from it. They negate experiencing their own body at the level of sensory feelings:

> *I didn't bother with the pain; I didn't speak much about it. It hurt like hell! (. . .) I didn't take painkillers, the doctors looked after me, but I felt good in my head, so I didn't feel pain. On the second day after surgery, I went to the students to take a look at me and drank coffee with them.* (Interview No. 2)

In the medical context, the donors are clearly dominated by a task-oriented perception and treatment of their body. They do not subject the body to deeper reflection, the body is not experienced by them in parameters other than medical. The donor's body is non-sensing and working. The donors focus on experiencing the recipient's body:

> *This pain is very similar to the pain in childbirth. If you see the recipient with the kidney later, it's like seeing a baby after delivery. When you see a person whose health is improving, it is such a joy as seeing a child after giving birth. The pain is forgotten.* (Interview No. 6)

The donors clearly remain more preoccupied with the recipient's physicality than the sensual sensation of their own body. A manifestation of the negation of their pain sensations is to avoid taking painkillers.

## 2. Changes in body appearance after donation

The change in the appearance of the body indicated by the donors as the first is the postoperative scar, which, however, according to the respondents' assurances, does not affect the well-being of the donor in relation to his body or his perception. Regardless of the method of kidney procurement (laparoscopic nephrectomy versus classic nephrectomy), the donors are not absorbed by the postoperative scar, nor do they focus their attention on this part of the body. In aesthetic terms, the donors distance themselves from it. The scar does not reduce the image aesthetics of the donor's body, but nor does it add to it. Instead, other functions are assigned to it. First of all, the presence of postoperative scars has an ennobling significance for the donors, especially in the context of male identity:

*I haven't even seen this scar. I have to look now. (. . .) Well. . . such a small thing remained, in one place it has almost disappeared. No, a guy doesn't mind having a scar, on the contrary, it ennobles him. Mr. Wołodyjowski had a scar on his face, and I have one on my stomach.* (Interview No. 5)

Secondly, the scar has a sentimental value for kidney donors, it is a positive stigma–a so-called bodily memento:

*The scar is a reminder of this event, it gives me the feeling that I have done something good. When I look at what I have on my stomach, when I look at these marks, I feel good.* (Interview No. 2)

Thirdly, the scar is for kidney donors a symbol of the existence of solidarity of the donor's and recipient's bodies since the fate of the recipient's body depended on the donor's body:

*For me it is positive, I look at this scar with sentiment. For me, it is a sign of this transplant. I approach this scar very sentimentally.* (Interview No. 3)

In interpreting the scar having sentimental value or being a symbol of solidarity of the donor's and recipient's bodies, the aspect of kinship or lack thereof is irrelevant. The cited definitions of the postoperative scar are typical and the type of bond between the donor and recipient does not differentiate them in any way.

Due to its location and size, the postoperative scar is not subjected to evaluation in public space; it remains an experience primarily of the donor himself. While the scar itself is not an aesthetic problem for donors, the abdomen as part of the body being the object of surgical intervention is experienced negatively in terms of image. This applies primarily to the surveyed women who describe their abdomen as distorted by donation. Irregularities occurring within it do not subside despite the passage of time, which is why the subjects miss the lost body form in this area. Here is an example statement:

*It has changed, my stomach is deformed. (. . .) It is not about the scar at all, but about the appearance of my belly. It's somehow deformed, shapeless because the fluid lingered a long time, the muscles were cut. (. . .) And so I always put on clothes that cover this defect. So, my*

*stomach suffered the most in terms of appearance–how do I look at myself as a woman*? (Interview No. 12)

Notwithstanding, the changes in the body appearance of kidney donors do not affect its presentation in everyday life. They are also irrelevant to the respondents' social activities and interactions. Donation does not narrow and does not interfere with body control in kidney donors on the plane of social interaction:

*When I say that I donated my kidney, everyone looks for the scar. "Well, where did you donate your kidney?" They look for a scar on my back, and I don't have one there. I had a laparoscopic collection; I have a small scar that my underwear covers. I don't have an aesthetic problem at all because even if someone knows, he can't find a scar.* (Interview No. 16)

The bodies of the respondents are considered socially adequate because they are similar in biological terms to other bodies in public space. Since they are in no way different from the modern canons of appearance, health and fitness, kidney donors are treated as full participants in social life. Therefore, donation and its bodily consequences do not affect the social identity of the donor.

## 3. Changes in body functions after donation

Kidney donors generally experience no changes in their body function after donation. If there are any functional limitations, they may be up to 2 months after the nephrectomy procedure. In the longer term, the donors strongly deny that they have any bodily limitations:

*Until now, my body has not given me any kind of sign that I did wrong, that my body lacks this kidney. It is still the body it was. Nothing has changed. I function just like I did before.* (Interview No. 16)

*I got tired faster for about 2 months after the surgery. (. . .) But now I'm back to normal.* (Interview No. 18)

The donor maintains control over his own body, and this means that his daily life is not disturbed in any way. Instead of dysfunction, the donors show some bodily benefits from the act of donation:

*I have no restrictions. It's just the opposite because thanks to kidney donation I got to know my body a little better and it's easier for me to control it. (. . .) I had to take care of myself to donate my kidney at all. I had to lose weight. And thanks to the fact that I gave this kidney, I lost weight. So, such a change in the body.* (Interview No. 5)

*Everything is fine. After donating my kidney, I rejuvenated. I feel younger and I look younger.* (Interview No. 19)

Better understanding of one's own body and its functional capabilities, greater skills in managing one's own body and better physical well-being are the most important benefits of donating the kidney which the donors perceive in the body dimension.

## 4. Donor's experience of lack of a kidney in the body

The lack of one kidney in the donor's body as a physical condition has no meaning for the studied donors. It is because this deficit in no way affects the functioning of the donor. The

donor does not experience the lack of one kidney at the physical level because he does not experience his body differently than when he functioned with two kidneys:

> *I do not pay attention to it, really. Only now, when I'm talking to you, I thought–oh yeah, I don't have a kidney on one side. It's not like losing an arm, something that is visible. Here I have no nerves of feeling like in my arm. So, I don't see that I don't have it and I don't feel that I don't have it. I don't feel that what is inside.* (Interview No. 18)

The donors do not reflect upon this condition because they do not feel the lack of one kidney at the level of sensory impressions; they do not experience it, e.g. through the sense of sight. The donors' statements deny that they have an emotional bond, or any other type of attachment associated with the donated organ:

> *It's so unimaginable to me that the kidney is working there inside my husband, but I'm not emotionally connected with it. It's good that it's with my husband, that it was of use.* (Interview No. 2)

The awareness that the body of the donor is lacking one kidney is linked with considerations of utility. The kidney given to the recipient is treated by the donor instrumentally, i.e. one that is not in the body of the donor as it turned out to be more needed for another body. The importance of kinship between the donor and the recipient does not emerge in this aspect of the study either. Treating the kidney in terms of a material existence of a useful character is common to both donors related to the recipient and those who remain strangers to each other.

## 5. Right of ownership of kidney given to recipient

The kidney donors undeniably admit that the kidney they donate is already the property of the recipient and the donor is not entitled to any rights of ownership. To justify their position, they present a broad argument that the kidney is the property of the recipient, regardless of the existence or absence of kinship between them.

And so, first of all, the kidney is physically located in and works for the recipient's body as a result of surgical fixation in his body, sealed by a medical institution as to its new location and destination:

> *It's already the recipient's kidney. I've never looked at it like it's my kidney. It is important that it is there and works well. I gave it away and it's not mine anymore.* (Interview No. 6)

Secondly, the kidney was physically absorbed in the recipient's body, was accepted by him, which proves its biological closeness and similarity to the recipient's body:

> *Of course, it's my son's kidney. The same blood type, maybe a little thinner veins, but it's his kidney. It doesn't matter if it's not where it should be. But it's important that it's connected to his body.* (Interview No. 24)

Thirdly, the donor no longer has a direct impact on its functioning, and this influence belongs only to the recipient:

> *I realized that it was no longer my kidney, this detachment took place. It's not mine anymore, it was given once and for all, and it belongs to him, that is my brother. The kidney is already his. (. . .) My brother realized that, as he once told me: "Your kidney works well for me." I*

*realized then that this kidney belongs to him and I no longer have the right to it.* (Interview No. 13)

Fourthly and finally, the donor cannot recover it because the act of donation is irreversible and reimplantation is impossible:

*It's now my son's kidney. How is it mine? It was mine! It's now my son's kidney. Should he give it back to me? No, it is not possible. I just donated my kidney and now it's his kidney.* (Interview No. 11)

## 6. Experiencing the kidney in the recipient's body

The donor, realizing that his kidney is in the recipient's body, thinks about it only in utilitarian categories. The reflection on the presence of the kidney in the recipient's body concerns the following: 1. how does the kidney function in the recipient's body? 2. how long will the kidney function in the recipient's body? 3. how to make the kidney function as long as possible in the recipient's body? Here are examples of the respondents' statements:

*Since the kidney has been accepted, it means that it is good there. That's how I always approach it. I didn't analyze it so deeply that my kidney is there. I focused more on its function, that it took over the functions.* (Interview No. 6)

*I'm happy that this kidney is with my son and that everything is fine. May it be good, and I believe it will be good. My son will take the tablets.* (Interview No. 11)

*I have no fondness for this kidney, but I would like it to work as long as possible for the recipient.* (Interview No. 18)

The awareness that the donor's kidney is in the recipient's body raises the need for the donor to control it, which should be understood as control over the recipient's body. This proves that the donor has an extended experience of his body because the limits of experiencing his own body have shifted and after transplantation extend to the recipient's body:

*I am glad that this kidney could be useful, that it works. In fact, I ask my husband everyday: "Have you drunk a lot today?" I want him to take care of this kidney, keep himself hydrated. I'm trying to control my kidney.* (Interview No. 19)

*I am very happy that the kidney works. When the tests come, I'm curious how the kidney will be working. If it turns out that something is wrong with the recipient, I would keep thinking that maybe I had a deficient kidney. But I think I took care of myself, that I filtered the kidney well and it would work well.* (Interview No. 21)

*It's nice that the kidney works. I am always happy that my brother, after tests, always calls me and lets me know how his kidney is working. My brother has great results. Like a young man with two kidneys. It is good that my brother shares this information with me.* (Interview No. 22)

Regardless of the existence/absence or type of kinship between the donor and the recipient, the kidney transferred to the recipient becomes an extension of the donor's body. The donor still feels responsible for its functions. The recipient himself confirms the donor's need for control by informing him of the results of kidney function after each check-up. Thus, sharing the

donor's biological individuality through the recipient's body on the one hand justifies the donor's belief that the transplanted kidney remains the recipient's property, and on the other explains the donor's expanded experience of his own corporeality, including the recipient's body.

## 7. Self-definitions and body perception in kidney donors

The body is primarily private property for the kidney donors. Nephrectomy appears as a conscious choice of the donor, as the realization of his rights of ownership to the body. Analysis of the motives and decision-making process of the donor supports the unequivocal statement that the donors, feeling themselves to be the owners of their bodies, decide themselves that they want to live in a body devoid of one kidney. The kidney donors are clearly aware of their body. They reflect on their own body and formulate numerous self-definitions. For the donor, what matters is not the state of the body, the incompleteness of its form, but above all its functions. This is an important category of survival of the body for the living kidney donors. It is not about the absence/presence of the kidney in the body, but about how the body functions, and the physical well-being of the donor. The kidney donors declare that they like their own body and accept it. Here are some examples of statements:

> *I don't have any problems with my body and accepting it, I didn't before the kidney was taken, and I don't now. Nothing has changed.* (Interview No. 5)

> *I like my body, although I could lose 5 kg because I have gained some weight lately, but it is cold, so it is known that fat accumulates. I will work on it.* (Interview No. 17)

Despite general satisfaction with their own body, the respondents remain aware of its imperfections, which, however, do not undermine the attitude of acceptance of the body, and are treated as a stimulus to work on it. And so, when it comes to the appearance of the body, the donor, even if he sees some disadvantages, justifies them and makes it easier to reconcile them. He also shows a clear reluctance to correct his own body in the field of aesthetic medicine. On the other hand, the donor's requirements and reservations are much higher for the functions of his own body. Body functions, in terms of fitness and body condition, are treated as a requirement the donor is willing to account for:

> *I try to run, I go to the swimming pool once a week. Once every two weeks I play football with my friends in a hall or in the open air.* (Interview No. 17)

The perception of one's own body after donation does not change in the studied kidney donors. The body is not seen as better or worse, healthier or sicker. It is not more perfect, but it is not weaker either. The body as a whole in the perception of donors is the same as before donation.

## 8. Changes in relation of kidney donor to his own body after donation

According to the respondents' reports, their attitude towards their own body changes after donation. The donors begin to consciously reflect upon and experience their corporeality more. They also feel more responsible for their body. The change in the attitude towards one's body noticed by the donors after nephrectomy consists in adopting a healthy lifestyle, including above all the tightening of dietary regimes, especially with regard to the amount of fluids drunk and greater self-discipline in physical activity:

*I care more about myself now. With drinking water, the body is different, so I drink water, keep my body hydrated. (. . .) For sure I take care, I train so that my stomach does not hang. (. . .) I bake bread myself; we don't buy bread. Only homemade salads with yogurt. I don't eat roasted meat, just cooked. (. . .) I pay more attention now to what I eat. My dinners are always steamed. I try to eat healthy food.* (Interview No. 24)

By becoming more aware of their body, the donors begin to observe it more closely and interpret the signals that flow from it. After donation, the donors also see a significant change in their attitude towards their own body in the context of its aesthetics. They begin to pay more attention to the appearance of the body and care about body image:

*For me, the body is something that someone will see first, before getting to know me. And if he sees me well, it will be easier for him to accept me and get to know me well. They judge me as they see me. I try to look good and present myself well and take care of my body. I don't exaggerate, but I don't neglect it.* (Interview No. 19)

The donors believe that the appearance of the body is perceived as a human business card. However, there is no narcissistic cultivation of the body's appearance or excessive concentration on its aesthetics. Although they admit that the appearance of the body is important to them, it is definitely secondary and still ahead of the value of body functionality. This category of experiencing one's body remains the most important for the kidney donors. The donors are not interested in aesthetic concentration on the body for the image issue alone. Here is an example of a statement:

*I eat healthy. I don't eat too much, I eat 5 times a day, I just came back from Nordic walking because I usually walk around 8 pm. I try to take care of myself. But I don't spend all my time at the beauty parlour. I think that taking care of yourself is not a visit to a beautician. I do not do that.* (Interview No. 16)

The donors are occupied by body aesthetics insofar as it does not interfere with the utilitarian treatment of the body and serves its functionality. What confirms both the conscious and utilitarian attitude towards one's own body is the declaration of consent of the respondents to posthumous organ donation, which is motivated by the desire to fully use their biological potential.

## Psychometric test results

The results of the psychometric tests were obtained on the basis of statistical analyses performed using the IBM SPSS Statistics 24 package. The level of significance of the observed relationships was considered as $p < 0.05$. The statistical tools used in the article included: analysis of differences between means (Student's t-test), the analysis of variance test (ANOVA), and Cronbach's alpha test. The selection of hypotheses for statistical verification was based on the principle of dialogue with problems raised during the sociological interview. This means studying statistical methods analogously to the sociological variables of body image.

## 1. Emotional attitude to donor's own body

In order to verify whether the respondents are characterized by strong negative or positive emotions in relation to their body, the obtained means were analyzed and compared to the sten scales. Analysis of the means shows an average level of assessment of their body in all the

respondents. This means there is no clearly positive or negative emotional attitude to their own body. The highest results interpreted on the sten scale were obtained by men in terms of Physical Attractiveness (M = 6.22), and the lowest by women in relation to Sexual Attractiveness (M = 5.67).

## 2. Body awareness

In order to verify whether the respondents may be characterized by limited awareness of their body, the number of responses *I have no feelings* regarding each of the 35 questions of the questionnaire was analyzed. It was assumed that this form of response may indicate limited access of consciousness to individual parts of the body. Limited awareness makes it difficult to make any (positive or negative) emotional evaluation. The analysis showed that less than 1/3 of the respondents' answers (only 24%) were characterized by a lack of emotional judgment. The analyses also showed similar results in the group of men and women. In the group of men 22% (M = 7.78), and in the group of women 26% (M = 9.25) indicated a lack of feelings towards individual elements of the body and its functions.

## 3. Perception of body parts most exposed to consequences of donation

In order to verify whether the parts of the body directly related to the surgery are assessed differently from the rest of the body image, a number of analyses of differences between the means were performed. At the beginning, three elements of the body image were identified, which were exposed to possible changes as a result of the surgery: the waist, hips, and the abdomen. Then it was analyzed whether the assessment of these body regions differs significantly from the others. Questions about the functions performed by the body were removed from the analyses. The results indicate a slight but lower assessment of these parts of the body in the entire group of respondents (Table 2). On the other hand, the division of the results by sex shows a statistically significant and worse (more emotionally negative) assessment of the abdomen in women (t(15) = 2.21; p <0.043) and the waist in men (t(8) = 3.21; p <0.012).

## 4. Body image consistency

In order to answer the question about body image integration, Cronbach's alpha analysis was performed on questions concerning the assessment of body parts. The analyses did not include questions dedicated to the functions performed by the body. The test values ranged from 0.93 (for women) to 0.96 (for men). At the same time, additional analyses showed that excluding any of the questions describing a specific part of the body would not disturb the strong picture of coherence. The results showed a very high correlation of all the responses with regard to the entire test, which may mean high body image coherence.

**Table 2. Differences in evaluation of waist, hips and stomach image as compared to other body parts in the whole sample.**

|  | N | M | SD | t | df | p |
|---|---|---|---|---|---|---|
| Waist evaluation | 25 | 3.08 | 1.32 | -2.55 | 24 | 0.018 |
| Evaluation of other body parts | 25 | 3.79 | 0.71 |  |  |  |
| Hip evaluation | 25 | 3.56 | 0.87 | -2.34 | 24 | 0.028 |
| Evaluation of other body parts | 25 | 3.79 | 0.71 |  |  |  |
| Stomach evaluation | 25 | 3.08 | 1.35 | -2.83 | 24 | 0.009 |
| Evaluation of other body parts | 25 | 3.79 | 0.71 |  |  |  |

## 5. Image-based and functional importance of donor's own body

The analysis of variance test (ANOVA) was performed in order to determine how important body image is for the respondents and how important the functionality of their own body is. The results show a significant effect of contrast of the main effects observed in the female group: $F$ (2.26) = 57.33, $p < 0.001$; $\eta^2 = 0.81$. The female test group attributes the highest rating to Sexual Attractiveness ($M = 44,73$; $SD = 7.65$) as compared to Weight Concern ($M = 31.65$; $SD = 6.81$) and Physical Condition ($M = 29.98$; $SD = 7.05$). The results also indicate a significant effect of contrast of the main effects observed in the male test group: F (2.16) = 45.604, $p < 0.001$; $\eta^2 = 0.85$. The male test group attributes a higher rating to Physical Condition ($M = 46.03$; $SD = 10.27$) than to Body Strength ($M = 30.59$; $SD = 5.54$). The male subjects also attribute a higher rating to Physical Attractiveness ($M = 38.61$; $SD = 6.89$) than to Body Strength ($M = 30.59$, $SD = 5.54$). The results show a differentiated body assessment depending on sex. It turns out that for women the image aspect is key–the questions related to sexual attractiveness. For men, the most important aspect is the functional aspect, and then second the image aspect–the questions about the physical condition of their own body, and second, its attractiveness.

## Conclusions

Both of the aims of the study have been achieved. As a result of carrying out the study, it was possible to develop a method consisting of complementary techniques: qualitative and complementary quantitative. The result of the employed procedure is complementary data revealing the multi-faceted nature of the identification of one's own body by living kidney donors. In response to the main research question, "How does interference with the human body, involving the removal of the kidney, affect the experience and body image of donors?", the following conclusions were drawn from the study.

### Way of experiencing the body by kidney donor

In the somatic dimension, the way kidney donors experience their own body is seamless. The routine of the donor's daily activities is not disturbed in any way, which proves that the body is fully in control. In the adaptive dimension, the kidney donors' experience of their own body determines the conscious and reflective experiencing of one's own body. Kidney donors show care and concern for their own body both in the aesthetic and functional dimensions. The bodily consequences of donation do not change the way the body is experienced in the social and interactive dimension of the donor's life. They do not in any way affect the shape, course, or results of donor interactions with other people. The way of experiencing one's own body in the identity dimension clearly indicates that the kidney donor's body is defined as the carrier of identity–the body as "I". The change in the donor's bodily identity after donation of the kidney is the expanded experience of one's own body, which also includes the recipient's body. The donors' experience of their own body and kidney in all the above dimensions is unrelated to the factor of kinship or non-kinship between the donor and the recipient.

### Kidney donor's body image

Kidney donors have a strongly integrated body image. The parts of the body that are directly related to the procedure (abdomen, waist) are rated worse than the rest of the body. A worse assessment means stronger, negative emotions related to the identification of a given body part. The obtained results also indicate that living kidney donors evaluate their bodies at an average level (no clearly positive or negative evaluations). For women, the most important

issue in assessing the body is its sexual attractiveness, while for men its functional aspect–physical condition–is most important.

## Discussion

The results of the psychometric tests are consistent with the data presented in the subject literature and indicate the lack of strong emotions accompanying the image of one's own body among kidney donors [50]. The literature also points to the important function of the perception and awareness of one's body in living organ donors. It turns out that they may have a protective effect on the subsequent risk of negative consequences after the procedure. [51]. The results of the presented studies also confirm an observed regularity–the smaller the surgical interference, the better the body image, including the areas directly related to the procedure [22, 24, 25]. The results of the authors' own research also refer to two effects observed in the literature: differences in the body image of donors depending on sex [52], as well as emotions as an element responsible for the consistency of the body image among this group of respondents [53].

However, the category of "experiencing one's own body" has not been subjected to research exploration in the context of living kidney donation. The sociological part of the presented research does not have any references in world literature, and therefore can be considered as pioneer. In the world literature, there are increasingly more publications in the field of medical sciences presenting research using grounded theory methodology [54–60], which proves that in this area there is a perceptible turn towards the individual and his subjective experience of such analytical categories as body, health and disease. Thanks to the use of the sociological version of grounded theory methodology in the study of the phenomenon of living kidney donation, it was possible to gain access to the subjective experiences of donors and to illustrate the process of giving new meaning to their own body and experiencing it in an aesthetic and functional dimension. The proper methodology of the grounded theory of the theoretical sampling procedure, which led to the inclusion in the research of subsequent groups of donors with different degrees of kinship with the recipient, and donors in no way related to the recipient, showed that the nature of the bond between the donor and the recipient is irrelevant to the way of experiencing one's own body after donation. This is because, whatever the kinship between the donor and recipient, the nature of the interference with the donor's body is the same. In the surveyed donors, a strong focus on the body as being material (the body as "flesh") is manifested, which is primarily due to the medical context of the study. Nephrectomy as a surgical procedure that significantly violates the body's integrity (after all, the donor is deprived of one kidney) makes the body present in the donor's experience as such. The kidney donors in the donation procedure become aware of their bodies. As healthy people with no health problems that would force them to deal with professionals and medical institutions, the donors generally experienced a lack of awareness of their body until nephrectomy [61]. A body that functions without problems and does not create restrictions remains hidden because it does not impose itself on the attention of its owner. The nephrectomy procedure changes the circumstances of the experience of one's own body, which ceases to be complete. In order for this bodily change not to endanger the sense of "I", donors must exercise effective control over the body. The concept of "I" is rooted in the experience of the corporeal because the body is treated as an obvious aspect of the person [20]. Therefore, the way donors experience their own corporeality, which is based on increasing their body awareness, is not influenced by the bond with the recipient, but by the interference in the body that redirects the donor's attention to it.

Anthony Giddens, one of the most revered sociologists of the twentieth century, points to the specific function that the body performs in modern times. The body is not only

*appearance*, but also *the way of being*, *sensuality*, and also the subject of various *regimes*. The above aspects also build our identity. They allow us not only to update our knowledge of ourselves in the perspective of passing time (age-related changes, diseases), but also in the context of social life (changes related to the concept of beauty or the social roles we play). The aforementioned author writes about *the reflective project of our Self*. [19]. The authors have also carried out this project due to conscious work on the above-mentioned aspects of corporality. The results of the presented research indicate that the experience of kidney donation may be conducive to raising the awareness of one's own body. The key question that arises in the light of the obtained data is about the possible mechanism of in-depth reflection of one's own body.

The answer to this question can be sought in the psychological processes of dealing with a difficult crisis situation such as donation. Most studies on the quality of life of donors after transplantation point to the fact that the mood of donors after the surgery periodically deteriorates [62]. Research also indicates that anxiety, apart from the risk of depression and grief, is one of the most serious problems faced by donors [63, 64]. It may have several sources, including anxiety about the functioning of the body after the donation [47], anxiety about the health of the recipient [65], expected difficulties in returning to the previous social roles [7] and the related deterioration of the financial status [65, 66]. However, data on the mental health of donors after donation show that, firstly, there is no increased risk of psychiatric disorders as a result of donation [2], and secondly, that most of the negative psychological consequences of donation disappear with time [62, 50]. This means that donors work out some form of coping with the stressful situation, which is the procedure and its consequences. The presented research may indicate that one of the coping methods is to temporarily focus one's attention, perception and thoughts on the body. The need for this reflection, as well as its depth, are obviously dependent on many factors that the study did not cover (e.g. the level of fear of donation, basic hope, the level of trust in the healthcare system, or social support). The body, however, seems to be a natural point of reference in the process of understanding the changes that follow donation. Nonetheless, the presented research goes beyond the psychological mechanism of coping. The qualitative results from the sociological interviews indicate rich sources of knowledge and experience that donors derive from the coping process. Giving meaning to the postoperative scar seems to be a perfect example. At the psychological level, this place is rated negatively (discomfort). In the reflective perspective and interpretation of the donor, the scar can be symbolic, sometimes ennobling.

## Research and application value of the project

The research procedure used in the project, a combination of grounded theory methodology and psychometry, proved to be effective (coherence of the obtained results). The obtained results indicate the possible areas of work in the long-term care of donors. This applies to both the diagnosis of the risk of adverse body image and development work. The first aspects should include, among others, defining the interpretation of postoperative changes seen in the body. The area of work that contributes to maintaining or increasing the quality of life is maintaining consistency in body image. The research also clearly indicates the need for high individuality of the approach to experiencing donations, as well as the diversity of this approach by donor sex. The body, which is the object of donation, is also an important source of information for donors themselves after surgery. Its image should also be a source of information for the medical staff involved in this process.

## Limitations of the study

The authors of the presented project are aware that the studies would increase their application value if they were prospective and the list of psychological variables verified in psychometry

had been larger. It should also be noted that among the respondents there was only one person experiencing actual (requiring a response from medical services) complications after the procedure. None of the respondents indicated any serious complications on the kidney recipient's side. Therefore, the study participants do not constitute a group representative of the living donor population. The obtained results represent a particularly important, albeit fragmentary, picture of the body experiences of a living kidney donor. This gives one the opportunity to return to research taking into account the perceived deficits.

## Supporting information

**S1 Questionnaire.**
(XLSX)

**S1 Appendix.**
(DOCX)

## Acknowledgments

We wish to wholeheartedly thank the kidney donors for their participation in the project and their openness during the interviews.

## Author Contributions

**Conceptualization:** Katarzyna Kowal.

**Data curation:** Katarzyna Kowal, Mateusz Zatorski.

**Formal analysis:** Katarzyna Kowal, Mateusz Zatorski.

**Investigation:** Katarzyna Kowal.

**Methodology:** Katarzyna Kowal, Mateusz Zatorski.

**Supervision:** Katarzyna Kowal, Artur Kwiatkowski.

**Writing – original draft:** Katarzyna Kowal, Mateusz Zatorski, Artur Kwiatkowski.

**Writing – review & editing:** Katarzyna Kowal.

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
