## [Decision Letter · Decision Letter 0]

7 Jul 2020

PONE-D-20-15970

Experiencing one's own corporeality and body image in living kidney donors - a report from interdisciplinary studies

PLOS ONE

Dear Dr. Kowal,

Thank you for submitting your manuscript to PLOS ONE. After careful consideration, we feel that it has merit but does not fully meet PLOS ONE’s publication criteria as it currently stands. Therefore, we invite you to submit a revised version of the manuscript that addresses the points raised during the review process.

We look forward to receiving your revised manuscript.

Kind regards,

Frank JMF Dor, M.D., Ph.D., FEBS, FRCS

Academic Editor

PLOS ONE

Journal Requirements:

2. Please provide additional details regarding participant consent. In the ethics statement in the Methods and online submission information, please ensure that you have specified what type of consent you obtained (for instance, written or verbal, and if verbal, how it was documented and witnessed).

Additional Editor Comments (if provided):

This MS is of interest, and would fill a gap in research relating to living donation. However, the reviewers have highlighted several issues with the paper and therefore it cannot be accepted in its current form. It would need to be rewritten addressing all comments of the reviewers, including revisions of the English used in the paper by a native speaker. Both reviewers have recommended a some parts of the paper to be shortened and more condensed (such as methods), and others to be eloborated on (discussion), and many parts need to be made more clear. Hopefully you can address the comments made by the reviewers in a point-by-point fashion and revise the manuscript accordingly. There would obviously no guarantee that a revised paper would be acceptable for publication, and will be thoroughly reviewed again.

Reviewers' comments:

Reviewer's Responses to Questions

**Comments to the Author**

1. Is the manuscript technically sound, and do the data support the conclusions?

Reviewer #1: No

Reviewer #2: Yes

2. Has the statistical analysis been performed appropriately and rigorously? 

Reviewer #1: I Don't Know

Reviewer #2: I Don't Know

3. Have the authors made all data underlying the findings in their manuscript fully available?

Reviewer #1: No

Reviewer #2: No

4. Is the manuscript presented in an intelligible fashion and written in standard English?

Reviewer #1: No

Reviewer #2: Yes

5. Review Comments to the Author

Reviewer #1: Abstract:

"Both the methodology used and the interdisciplinary nature of the research are definitely innovative" - This needs to be moved to the conclusions as it is a statement about why the authors feel their paper is contributing to the literature

Whilst the sociological perspective is not common within this field, it is by no means unique and has been incorporated into other studies.

General comments:

The English throughout the paper (and in particular the introduction) is not of the standard expected for publication in an international journal. The authors need to work on the paper considerably to improve this.

I was unable to access the data stated to be within a dropbox folder as the link was invalid. If you feel this data (or some of it) is relevant then it ought to be placed within a supplementary file accompanying the paper online.

The paper feels very long and I have made some suggestions below to bring it down

I think the findings are of interest but the conclusions reached are possibly too focussed on body image; inadvertently addressing the other factors that are interplayed within this complex group of patients.

Introduction:

Page 3: Second paragraph, last sentence: the word ‘his’ should be changed to ‘their’ in order to make this gender neutral

Page 4, paragraph 2: I was very unclear what you were trying to say in the second half of this paragraph. I don’t believe you have searched for a new methodology – you have possibly attempted to approach the topic of body image by utilising mixed methods or a qualitative approach, but this is not really a new methodology.

Page 4: third paragraph, second sentence: You say ‘magazines’, do you mean journals?

‘diagnose the consequences of donation’ – I’m not sure this is phrased correctly.

The second half of this paragraph again slips into the remit of the discussion. You need to justify why you chose this method rather than state what it allowed you to do as part of your analysis. Again this paragraph is poorly written and needs to be completely rewritten.

Page 4/5: When stating the study aims you need to be clear at the beginning by saying ‘the first aim of the study was x, the second aim was y’. ‘meeting the second purpose of the study’ is the wrong phraseology.

Page 5, paragraph 2: ‘The psychometric study provided a different kind of data’ – again you need to say that this is a mixed methods study with a qualitative and quantitative component and you need to justify why you have adopted this approach, and which part provides which data.

The final sentence on page 5 makes no sense. I would say that these different areas overlap, rather than them being bordered by each other.

Methods:

You need to state that this is a mixed methods study incorporating qualitative interviews analysed by grounded theory and a questionnaire which has provided quantitative data. Again, this methodology is less common in this field but this study is not unique in utilising this approach

Table 1: Employment status – please say ‘employed / unemployed’ as active / passive is unclear

I don’t think it is necessary to detail the entirety of the donation operation. A simple explanation would suffice and may benefit from a diagram if you wish to show where the scars are located.

Page 9:

Examination procedure: I am unsure what ‘no masking instructions’ means.

Page 10:

Under qualitative methods – I would not call this ‘sociological’ as it is not unique to sociology.

I think ‘intensive’ interview should perhaps be an ‘in-depth’ interview as this is how we would refer to it in English. The word ‘intensive’ sounds more aggressive than an in-depth interview, which as you say aims to thoroughly examine the views of the participant through focussed conversation.

Page 11:

Final paragraph: How did you reach the conclusion that donors opened up more during telephone interviews rather than in person? This has not been the experience of a lot of participants in other studies to my knowledge. Also, you are not able to guarantee that the interview has been conducted in complete privacy as you are unclear as to who else is close by to the interviewee at the time of the interview. I know you mention that you asked the participants to consider these issues but do you have a guarantee that the optimum conditions were met?

I also understand the desire not to have patients attend the hospital but I am unclear how conducting it in their home deprives the study of confidentiality. Please can you explain?

The description of both the qualitative and quantitative methods is way too long and could be better summarised.

Results:

These are of interest due to the detail obtained through the qualitative interviews.

The results of the quantitative component, and the specific analysis of male vs. female views, is also very interesting.

I like the quotes being included within the text but it does make the whole paper very long. One strategy may be to place these in a table so that they don’t distract from the themes presented. Either that or if they could be made shorter that would also help.

Conclusions:

These need to be included in the first part of the discussion. I would remove the numbers from these paragraphs. They are not needed. I would also attempt to rewrite these paragraphs in order to be more concise with your statements and to bring the word count of the paper down and so they provide a short summary prior to the more detailed discussion

Page 29, paragraph 3: You mention that anxiety is a basic problem faced by donors. I would phrase this as it being a regular problem as basic implies that it is minor. I would also argue that you cannot conclude that it is primarily related to their body. There are a number of anxieties experienced by donors, primarily often related to their recipient.

You have also misquoted reference 62. This paper did not demonstrate that negative psychosocial difference disappeared over time. It showed that very little changed after donation when it came to a number of psychological factors.

Reviewer #2: I was really looking forward to reviewing and reading this paper, as it is indeed an aspect that has not been studied thoroughly or at least in this way so far and that interests me personally, as I have been working with donors and conducting research on living donation for several years.

I definitely enjoyed reading this paper. Its strengths are

1. that it is one of the few studies addressing this issue

2. the novelty of the approach in using mixed methods to address the research question

3. that it describes thoroughly (at least in most parts) the methodology offering high levels of transparency

4. the obvious dedication of the researcher to the study subject.

I would want to see this paper published, yet there are few minor or maybe not so minor aspects I would like to comment on and that I think need to be addressed before publishing. Please understand my comments not as criticism, as I think this is a very interesting study, but rather as an exchange of opinions and ideas you could address for potentially improving the paper.

1. Regarding the title: I think the paper would benefit from slightly improving the title, as I feel that the second part of it “a report from interdisciplinary studies” does not reflect exactly the content of the paper. It gives to me at least the impression it is not an original study but a review; secondly the paper in my opinion describes complementary methods or mixed methods used to address the research question and appears as one study/project, so the plural „studies“ is confusing

2. In the abstract, in results, the wording „reduced assessement“ feels unclear.

3. Line 76 „studies indicate no change in the overall quality of life“: this is an oversimplification and not a fair representation of the studies so far regarding QoL in LKD.

4. lines 90-92: the meaning of this phrase is not clear to me

5. The research questions is stated clearly at the end of the paper 832-834, but I think it would be good to mentioned so clearly also in the beginning where you state the aims of the study 146-151, as I think it will help the reader

6. I am not sure I understand why the aim of the study is to search for effective and interdisciplinary research methods, (148-149) as you basically are already using them.

7. Table 1. It would make sense tob e more explicit regarding the donor-recipient relationship under related, unrelated (is it parent to child, spousesm friends? etc.)

8. I understand it might be important to mention the types of surgery laparoscopic or open nephrectomy, but I find it exaggerated for this type of paper. It would make more sense to explain to the non-surgery affiliated reader the bodily implications or risks of these two types of nephrectomy in order to understand the impact of each, which I suppose is one of your aims.

9. The same goes for some other parts oft he paper that I consider to be overly detailed, e.g. „the postoperative care of the donor“ 231-239 lines, 255-265,

10. Line 245: so who are these persons who carried out the study??

11. the psychometric data were collected 6 months later. This is a problem and it needs tob e justified why and to state the limitations or strengths this approach and the time gap means

12. this comment applies tot he whole paper but reading lines 286-290 made me ask myself even more about it, why is this study or at least the qualitative part a sociological one? The dimensions you include in the interview outline lines 286-290 seem more psychological to me . The same applies later on for the results, I have difficulties identifying the sociological character of it when speaking about coping, adaptation, perception, terms that in my experience at least belong more to the field of psychology. I understand sometimes boarders between disciplines are not clear, and I could place the study in the field of social sciences in general and not sociology necessarily.

Connected to this, I understand the study more as a mixed methods approach study and less interdisciplinary in this way, but you can argue about that.

13. The methods are described in a very detailed manner. This increases understanding and transparency but it feel sometimes a more condesned presentation would leave more space for more discussion.

14. I do not understand line 339.

15. The interview outline is missing and I think it should be added in a table or annex.

16. Lines 420-422. Who collected the psychometric data?

17. My major comment regarding this paper has to do with the presentation of the results in connection with the theoretical sampling you mentioned you applied.

You say you applied theoretical sampling and included in your sample non related donors and donor from paired exchange donation. Yet the results do not mention anything related to this aspect, whether it makes a difference or not. I think this should be addressed. Does it not make any difference and if not why? in the discussion

Furthermore, it is a pity you could not include during your theoretical sampling more different or extreme cases of donors who experienced complications, or their kidney was rejected or the recipient died or had complications. This makes the results appear a bit flat and less complex as they might be, and they „beautify“ or simplify the experience, as the way the donors experience their body and body image is embedded and it depends possibly on the different relationships and contexts and postoperative courses, complications etc.

There are reports of donors experiencing fatigue after donation which is the opposite oft he highly functioning body you mention. It would have been interesting to include such donors in your sample to broaden the „grounded theory“ derived from the data.

I understand this is not possible now and maybe even not the point. Yet, in this case I think you should mention all these aspects in the discussion and in a part called limitations of the study. Otherwise there is the danger that your results are adopted uncritically by clinicians or others to promote specific interests, as your results reflect only a part oft the phenomenon living kidney donation and this should be made clear that this is a sample of successful donations

18. It would help to add in annex the questionnaire you applied, to help the reader understand the psychometric part of the study and the result better.

19. unfortunately the study is not prospective so the statements of the donors about their attitude towards their body before donation is difficult to acceopt.

20. The hypothesis you developed based on the qualitative part and tried to test through the quantitative are not very clear. It think the part lines 767-824 needs to be presented more clearly.

21. Also when you say that some body parts are rated lower it is not clear to me exactly what lower means. e.g. line 857

22. I find the interpretations of the results in the discussion about focus on health behaviour and body care very insightful! and useful.

6. PLOS authors have the option to publish the peer review history of their article (what does this mean?). If published, this will include your full peer review and any attached files.

Reviewer #1: No

Reviewer #2: No

---

## [Author Response · Author response to Decision Letter 0]

6 Nov 2020

The article was subject to editorial and reviewers' evaluation, which was received by the authors of the text on 7th July 2020. 

The response to the reviewers' comments was included in the rebuttal letter as well as in the body of the article itself. The corrections made to the content of the article also include replacing the original version of the article title ("Experiencing one's own corporeality and body image in living kidney donors – a report from interdisciplinary studies”) given at the beginning of the cover letter. 

Thank you for your thorough and substantive evaluation of our article. Thank you for the received comments; we were able to make significant corrections in the text. This applies to both the level of the language used and the substantive assumptions. 

I convey the authors’ answers to the Reviewers’ comments. We wanted our responses to be as precise as the Reviewers' comments, therefore, the responses have been presented in the form of a table containing references to specific comments of the respective Reviewers.

Response to comments from Reviewer No. 1

1. and 2. Thank you for your comments on the content of the abstract. They all contributed to its present, much more perfect version.

3. Thank you for this remark. An editor specializing in scientific translations worked on the linguistic layer of the text.

4. We are sorry for this situation. Attached, as a file of supplementary information, we have presented an anonymized set of quantitative data (in the form of Exell database) collected in the psychometric part of the research, in order to potentially replicate the research results.

5. The article has been shortened in all the places indicated by the reviewers. Thank you very much for this remark.

6. The conclusions were divided into two parts, which correspond to the scope of the research issues in the sociological (qualitative) and psychological (quantitative) part of the research project. Therefore, in the first part of the conclusions we presented the conclusions from the research on the way of experiencing their own body by kidney donors, and then in the second part of the conclusions about body image.

7. Thank you for pointing out the linguistic errors. They have all been improved. 

8. Thank you for this extremely substantive remark. We fully agree that our methodology was not innovative, but simply mixed.

9. Thank you for pointing out the linguistic errors. They have all been improved. 

10. Thank you for this remark. We agree with the incorrectness of this wording, which has been corrected.

11. Our analytical "going forward" instead of focusing on the motives for choosing the grounded theory methodology resulted from the fact that we wrote this part of the text at the end. The Reviewer's remark is correct, and we fully agree with it. Amendment consisting in redrafting was introduced.

12. We made corrections according to the Reviewer's suggestions. They concern not only the form (changing the place of presenting the research question and research goals), but also the content (clarification of the first research goal). We admit that the Reviewer's comment was very valuable and allowed us to increase the precision of the description and obtain greater transparency of the Introduction. Thank you for pointing out the phraseological error. It has been removed from the content.

13. In line with the Reviewer's remark, we emphasized once again the mixed nature of our methodology, explained the reasons for adopting such a methodological approach and indicated which part of the study provides which data.

14. Thank you for this remark. This unfortunate sentence has been removed.

15. In the text, we have repeatedly emphasized the mixed nature of the methodology used in the research and removed the content suggesting the unique nature of our research.

16. We made the suggested changes to the table regarding the employment status of the surveyed donors. Thank you for this comment.

17. Thank you for this remark. We have resigned from the detailed description of the kidney donation procedure in favor of a simple and concise explanation of what this procedure is about, the differences between the two methods of kidney donation as well as the anatomical and physiological consequences of donation.

18. In response to the reported difficulty in interpreting the concept of "no masking instruction", as well as the Editor's comments, the part of the text entitled "Examination Procedure" was supplemented with a full description of the steps leading to informed consent to participate in the study. We hope that this will increase the transparency and show the importance we attach to the ethical side of research.

19. Firstly, we resigned from the title of "Qualitative Methods - Sociological ", which in its unfortunate wording could suggest that qualitative methods are equated by authors with sociological ones and should be treated as such. We are aware that qualitative methods are not unique to sociology and are also used by representatives of other humanities and social disciplines such as psychology, pedagogy or ethnology. However, this does not change the fact that the first author of the text and at the same time the researcher of this part of the research project conducted sociological research because she consistently used the methodology of the theory in its sociological version, reaching for the interpretative social constructivism of Kathy Charmaz, who draws her methodological inspirations from pragmatism and symbolic interactionism (humanistic sociology, alternative understanding) and phenomenology.

20. We agree with this remark. Since the intensive interview as a method of collecting data in the constructivist variant of K. Charmaz's grounded theory methodology is essentially the same as the in-depth interview, we opted for the consistent use of the term “in-depth interview”. In fact, it is used more often in both English and Polish literature on the subject. We also explained in the text why we had used the term "intensive interview" when describing the method of data collection. We considered this explanation indispensable.

21. In response to the Reviewer's question about the findings regarding greater openness of the respondents during the interviews conducted by telephone, we would like to emphasize once again the fact of conducting the pilot research. It is on its basis that we established that in-depth interviews conducted by contact via telephone contain more data being a so-called dense description. The narratives of the subjects were more detailed than when the researcher met the subjects face to face. According to the researcher, this was primarily because the subject of the interview focused on kidney donation as an intimate life experience. Based on the comparison of the rhythm and content of the conversations with kidney donors made over the phone and in person, the researcher concluded that the telephone conversation gives the respondents a greater sense of psychological comfort, which results in greater freedom of expression. Without seeing the researcher, the kidney donors did not feel embarrassed to share sometimes very intimate details about their bodies. Of course, the researcher realized that in communicating with kidney donors via telephone he loses the possibility of observing their non-verbal speech, which according to A. Mehrabian's rule constitutes 555 of communication. In fact, during the telephone conversation, the researcher did not notice the respondents' body language, although she did feel it at many times. We would also like to strongly emphasize that the interviews carried out in the sociological part of the research were not telephone interviews (CATI), but in-depth interviews (IDI) conducted via the telephone. This formula of in-depth interviews was also determined by the high costs of direct interviews. Due to the large dispersion of the study participants, the costs of reaching each of them were impossible to cover by the research team. We did not have any grant for this research project.

Of course, the researcher could not be absolutely sure that no other people were present during the interview. However, she became more certain on listening to the content and course of the conversation (also while listening to the recordings). The researcher also referred to her own intuition and counted on the elementary honesty of the respondents.

22. Conducting the study at the respondent's home significantly violates his privacy because the researcher knows the home address of the research participants (sensitive data), which is contrary to the principle of research anonymity.

23. The description of the qualitative methods has been shortened so that the contents remain in line with the criteria for reporting qualitative research on the COREQ list recommended by PLOS ONE:

https://journals.plos.org/plosone/s/submission-guidelines#loc-qualitative-research

https://academic.oup.com/intqhc/article/19/6/349/1791966

24. We focused once again on the quoted statements of the respondents. In those places where shortenings could have taken place without sacrificing text quality, we have eliminated some quotes. Thank you for this suggestion. Nevertheless, we could not include the quotes in a table because in our opinion this would make reading the manuscript much more difficult as it would mean the constant need to distract from the research results and refer to a table in which the reader would be forced to find the relevant fragment of the respondent's statement.

25. 26. 27. As suggested by Reviewer, 1 the form and content of the Conclusion part have been revised. Nonetheless, we would like to separate it from the Discussion and present a short summary of the research results based solely on the obtained data. We also would like to present this information in two separate perspectives: qualitative, concerning the experience of the body, and quantitative, concerning its image. This reflects the adopted research methodology. In the Discussion part, based on information from the literature on the topic, we integrate two types of data and try to interpret them. It is a deliberate procedure, also reflecting the interdisciplinary nature of the research and our cooperation. The surgeon, body sociologist and clinical psychologist worked on the project in concert but based on research tools specific to their field. The meeting, summary and reflection on the effects of the joint work are included in the Discussion.

28. The remark about the anxiety experienced by the kidney donors seemed especially important to us. Indeed, narrowing the psychological problems faced by donors solely to anxiety is a significant simplification, even more so, ascribing fear only to matters of one's own physicality. Therefore, the content of the article was changed. In response to this remark from Reviewer 1, the source data was also supplemented with three new references.

29. Thank you for detecting inaccuracies in citing the publication sources. We have read the content of the source article once again. The effect of this is its removal from this part of the Discussion and the insertion of sources corresponding to the content.

Response to comments from Reviewer No. 2

1. Thank you for your suggestion to change the title. We trust that the current version more fully reflects its content.

2. Thank you for your comment. This information has been clarified.

3. We strongly agree with this remark. The indicated sentence has been changed into a more complete statement, supported by additional publications.

4. Re-reading the part of the introduction that was criticized by the Reviewer indicated that: a. the sentence was in fact unclear; b. its removal does not change the main idea of this part of the work.

5. The main research question was included in the Introduction section. Thank you for this comment. In fact, the formulation of the main research problem at this point allows us to follow our search for the theoretical foundations of the research (including the selection of publication sources), identify the specific objectives of the research, and also bring closer the decision regarding the selected research procedures. The question has been inserted before the detailed description of the objectives to make the text clearer.

6. Thank you for this extremely valuable comment. We have clarified the first research goal, which reflects our actual research intentions.

7. We explained what kind of relationship exists between the donor and the recipient in the study group. We supplemented this data with numbers.

8. Thank you for this remark. We have resigned from the detailed description of the kidney donation procedure in favor of a simple and concise explanation of what this procedure is about, the differences between the two methods of kidney donation as well as the anatomical and physiological consequences of donation.

9. In the 2 fragments of the text indicated by the Reviewer, we have shortened the content so that it is not too detailed.

10. We supplemented the part describing the examination procedure with information about who conducted the research.

11. In response to the comment from Reviewer 2, we supplemented the content of the article in the part devoted to the examination procedure. We treat keeping the time gap between the sociological and psychometric parts of the research as a deliberate procedure, motivated by the need to maintain the separateness of two types of data: a. Qualitative data on experiencing one's own body; b. quantitative about his image. Conducting the research simultaneously could result in interaction of the conveyed content, e.g. through the occurrence of the priming effect. The interval between the quantitative and the qualitative part of the study was set at approximately 6 months. This time was assessed as optimal for the expiration of the memory trace of most of the interview content, and at the same time it did not significantly change the identification of one's body.

12. In response to comment 12 from Reviewer 2, allow me to enter into a discussion. The presented research, in its sociological part, is sociological research, as evidenced by:

1. The research methodology - the researcher consistently uses the methodology of the grounded theory in its sociological version (Kathy Charmaz's social constructivism).

2. The theoretical framework of the conducted research, which is based on such sociological theories as symbolic interactionism (Anselm L. Strauss) and social phenomenology (Maurice Merleau-Ponty, Alfred Schütz).

3. The interpretation of the research results using sociological categories and concepts specific to the language of the discipline.

For the above-mentioned reasons, the presented sociological research cannot be called psychological, nor can it be generally presented as research in the field of social sciences.

The dimensions of sociological research on the experience of the body by kidney donors (1. somatic; 2. adaptive; 3. socio-interactive; 4. identity) presented in the text are obviously not those issues that can only be investigated using sociological methods. I agree that they can also be successfully researched by a psychologist. Nevertheless, a sociologist will examine and describe them differently than a psychologist. They will reach for different research methods, but they will also use different frames of reference, which are the resources of theoretical knowledge specific to these disciplines. The sociologist carrying out this part of the research, being a qualitative sociologist, assumed the role of a meaning seeker and focused on understanding the meaning of nephrectomy in the context of experiencing one’s own body by the donors and the impact of this interference with the body on the self-conceptions and personal identity of the donor kidney. In exploring the identity consequences of donation, the researcher, as a sociologist, focused on the socio-interactive embedding of an individual's identity.

Our research, in its sociological part, is not pioneering research on the body in social aspects. There is a whole separate field of sociology called sociology of the body that arose in the mid-1980s. The sociology of the body is a sociological subdiscipline with a well-established cognitive and scientific identity, as evidenced by the multitude of scientific works published in this field. As an example, I will give the textbook The Body and Society. Exploration in Social Theory (1984) by Bryan S. Turner, who is considered to be the founder of this field of sociology, and the peer-reviewed journal Body & Society, published since 1995 (editor Mike Featherstone), which publishes texts on key topics in the field of body research. Common to theoreticians and researchers in the field of the sociology of the body is the thesis that human identity is rooted in the physical sphere. Therefore, all the above-mentioned dimensions of researching the experience of one's own corporeality have been and are studied by body sociologists and medical sociologists (especially in the interpretative trend), as exemplified by numerous works devoted to the relationship of the body and identity in the context of health, disease or the development of medical technologies interfering with the body. Finally, I will recall examples of such works:

Charmaz K. Loss of Self: A Fundamental Form of Suffering in the Chronically ill. Sociology of Health and Illness. 1983; 5: 168-195.

Charmaz K. The Body, Identity, and Self: Adapting to Impairment. Sociological Quarterly. 1995; 36: 657-680.

Charmaz K. Good Days, Bad Days. The Self in Chronic Illness and Time. New Jersey: Rutgers University Press; 1997.

Nettleton S. The Sociology Health and Illness. Cambridge: Polity Press; 2006.

Nettleton S. „I Just Want Permission To Be Ill”: Towards a Sociology of Medically Unexplained Symptoms. Social Science and Medicine. 2006; 62: 1167-1178. 

Yoshida K. Reshaping of Self: A Pendular Reconstruction of Self and Identity Among Adults with Traumatic Spinal Cord Injury. Sociology of Health and Illness. 1993; 15: 217-245. 

13. The description of the qualitative methods has been shortened so that the contents remain in line with the criteria for reporting qualitative research on the COREQ list recommended by PLOS ONE:

https://journals.plos.org/plosone/s/submission-guidelines#loc-qualitative-research

https://academic.oup.com/intqhc/article/19/6/349/1791966

14. We explained that in writing "naturalized transcription" we mean a copy of spoken discourse.

15. As the mere outline of the interview might not be sufficient, in Appendix 1 we have presented a complete list of the researcher's information needs, which includes both general and open-ended questions as well as specific and structured questions the respondents were asked in order to explore particular threads of the conversation. We decided that such a formula would allow the Reviewers and Readers to get to know the interview scenario better.

16. Information about the person carrying out the psychometric tests has been supplemented in the text. Thank you for noticing this important deficit.

17. A. Thank you for your remark on the importance of kinship in the way donors experience their own body. In the presentation of the results and in the conclusions from the research, we wrote that the relationship of kinship was irrelevant to individual aspects of the donors' experience of their own corporeality. The analysis of each of the research threads confirmed the lack of importance of the kinship between the donor and recipient for the way the subjects experience their own body. Furthermore, the discussion was expanded to explain why kinship is irrelevant to the way one's body experiences after donation.

B. We consider this remark extremely valuable in the context of further sociological research on the experience of their own body by kidney donors. Notwithstanding, this will require a reformulation of the research project and obtaining a separate consent from the bioethics committee for the inclusion of donors with complications or those whose kidneys were rejected by the recipient's organism or the recipient died. Such an extension of the study group will certainly enrich the theory generated from the research conducted so far. Thank you very much for this remark, which has inspired us to carry out further research projects on the phenomenon of living donation.

C. We consider the comment to be one of the most important and key changes in the text. In fact, the tone at the end of our article would only indicate positive donor experiences. We agree (based on the extensive literature on the subject and our other research) that this is not true. We strongly agree that we could obtain completely different results in groups experiencing significant post-donation difficulties (both their own and the recipients). Therefore, in the part of the article devoted to research limitations, we made a change. Hopefully, this will be a clear indication that no hasty conclusions can be drawn from the obtained results. The article is absolutely no voice in the "for versus against living donation" discussion. It presents facts, and the observance of scientific rigors and the transparency of the results allows it to be treated as an important voice in the scientific discussion.

We also wanted to explain the effect of the lack of people who experience serious consequences of donation (their own or indirect recipients) in the group of respondents.

The subjects were recruited to the project in accordance with the rigors and principles of the grounded theory method. Until the theoretical sampling method was applied, as a result of which donors unrelated to the recipient were included in the study, the properties of the central category of the study (experiencing one's own corporeality) were uniform. This prompted the researcher to ask whether this category could be differentiated by the lack of kinship between the donor and recipient. Following the logic of theoretical sampling, the researcher sought here to discover differences within the analyzed analytical category, i.e. within the process of experiencing their own body by donors. Although this direction of analyses did not confirm the researcher's assumptions, it significantly expanded the analysis, in particular, by including unrelated donors in the research, another important analytical category was developed that had already emerged from the theory, which was "extended bodily identity". It turned out that donors experience a broadened body identity also when the recipient is a stranger to them. Thus, it is not the kinship that causes this state, but the effective control of the body, the meaning of which is to maintain the unity of the bodily identity. To sum up, the failure to include donors experiencing their own complications or the recipient's complications in the study group was a consequence of the direction of analytical work chosen by the researcher.

We are aware that such a selection of the studied group limits the possibility of a full answer to the research question, narrowing down the picture to well-functioning donors. The results of our research to date allow us to extend the research to include groups of donors indicated by the Reviewer. This is our intention in the near future. This project, as already mentioned, will, however, require significant changes. Firstly, it should take into account an additional variable, which will be medical complications or psychological trauma. Secondly, with such a sample selection, we face logistical challenges, including obtaining the necessary consent of the respondents and not interfering with the procedures of the study in the process of treating patients. Third, we need to obtain the Ethics Committee's approval again to research a group of donors experiencing complications directly or indirectly. Fourthly, the technique of conducting in-depth interviews must be changed because the interview with a traumatized person should be additionally secured with psychological support.

18. The Body Esteem Scale, adapted by Małgorzata Lipowska and Mariusz Lipowski, is a tool open to the public in Poland. Nevertheless, we do not have permission to make its original version available in scientific publications. In the text of the article, though, we provide the publication source regarding the questionnaire. It describes not only all the questions of the questionnaire, but also the entire procedure of adapting this tool. In accordance with the principle of transparency, I am sending the database on which the statistical analyses were performed. The database contains a record of all the questions in the questionnaire. Responding to the reviewers' comments, we added sample questions to the description of the tool for each subscale, as well as for both sexes.

19. Thank you for this comment. We agree that our research is not prospective. Therefore, in response to this remark, we have removed from the research results, conclusions and abstracts all those passages that pertain to how donors experienced their own body before donation.

20. Thank you for pointing out the lack of clarity of the information provided. The descriptions of the hypotheses and analyses have been corrected.

21. Thank you for this remark. In fact, merely identifying a “worse” assessment of the body parts directly related to the procedure may not be sufficient. Therefore, both in the abstract and in the section devoted to the conclusions, this information has been clarified. A poorer rating means receiving results that indicate a negative emotional evaluation of the body part in question.

We hope that the above answers correspond to all the comments submitted by the reviewers.

---

## [Decision Letter · Decision Letter 1]

9 Dec 2020

PONE-D-20-15970R1

Experiencing one's own body and body image in living kidney donors – a sociological and psychological study

PLOS ONE

Dear Dr. Kowal,

Thank you for submitting your manuscript to PLOS ONE. After careful consideration, we feel that it has merit but does not fully meet PLOS ONE’s publication criteria as it currently stands. Therefore, we invite you to submit a revised version of the manuscript that addresses the points raised during the review process.

ACADEMIC EDITOR:

I'm very happy to see the extensive revisions to the paper. The MS has significantly improved. As highlighted by both expert reviewers, it would still need a bit of work to make it acceptable for publication. It needs further shortening to avoid repetition/redudancy, and corrections by a native English speaker would be necessary.

Please follow the advice by the reviewers for the (hopefully) final revision;) Your work would definitely be interesting to publish.

We look forward to receiving your revised manuscript.

Kind regards,

Frank JMF Dor, M.D., Ph.D., FEBS, FRCS

Academic Editor

PLOS ONE

Additional Editor Comments (if provided):

I'm very happy to see the extensive revisions to the paper, which has significantly improved. As highlighted by both expert reviewers, it would still need a bit of work to make it acceptable for publication. It needs further shortening to avoid repetition/redudancy, and corrections by a native English speaker would be necessary.

Please follow the advice by the reviewers for the (hopefully) final revision;) Your work would definitely be interesting to publish.

Reviewers' comments:

Reviewer's Responses to Questions

**Comments to the Author**

1. If the authors have adequately addressed your comments raised in a previous round of review and you feel that this manuscript is now acceptable for publication, you may indicate that here to bypass the “Comments to the Author” section, enter your conflict of interest statement in the “Confidential to Editor” section, and submit your "Accept" recommendation.

Reviewer #1: All comments have been addressed

Reviewer #2: All comments have been addressed

2. Is the manuscript technically sound, and do the data support the conclusions?

Reviewer #1: Yes

Reviewer #2: (No Response)

3. Has the statistical analysis been performed appropriately and rigorously? 

Reviewer #1: Yes

Reviewer #2: I Don't Know

4. Have the authors made all data underlying the findings in their manuscript fully available?

Reviewer #1: Yes

Reviewer #2: Yes

5. Is the manuscript presented in an intelligible fashion and written in standard English?

Reviewer #1: Yes

Reviewer #2: Yes

6. Review Comments to the Author

Reviewer #1: Thank you for resubmitting your article for review, which has undergone extensive revisions. It reads considerably better now. I have a few minor comments:

Abstract:

Results

2. “lack of kidney in the body” would read better as “absence of a kidney in the body”

Conclusions – final sentence is very wordy. May read better as: The proposed approach utilising mixed methodology within the fields of sociology and psychology for researching the phenomenon of living kidney donation is definitely innovative.

MAIN TEXT:

Line 74 – “… negligible risk of physical and mental health disorders for living kidney donors”

Line 170 – The first concern that needs to be taken into account is the highly subjective…

Table 1:

Please clarify what is meant by “permanent / no permanent relationship”

Line 213: should read “symphysis pubis”

Line 214: shorter pain is incorrect. Should read ‘pain is less’ or ‘pain is reduced’

Line 237: consent being obtained ‘verbally’ is better English

Line 320: should read “were more likely” (past tense)

Line 333: Would read better as ‘None of the invited participants declined to participate in the interview, nor ended it prematurely’

Conclusions:

Rather than use the word ‘goals’ I would refer back to the aims for the study – ‘Both of the aims of the study have been / were achieved’

I’m unsure what ‘project realization’ is

Reviewer #2: I think the comments of the reviewers have been addressed in an appropriate and extremely thorough manner. The paper feels much more concise and accurate now, it is also shorter and the the findings are more visible as well as their importance.

Personally, I might have a different opinion on the theoretical discussion on what is a sociological and what a psychological study, but I respect the authors´view and thorough argumentation, so I will not oppose to this.

Still there are a few concerns in my opinion regarding the paper that can and need to be addressed

1. the paper still feels too long and there is a lot of repetition mostly when it comes to the description of the method (especially grounded theory and the sociological background of the author). As much as I admire the passion of the first author regarding the methods, they way it is written makes it at least for me difficult to follow sometimes due to repetition, on expense of the results. Thus, I would recommend the authors identify repetitions and re-write in a more condensed manner.

2. Even though I am not a native english speaker, I still think there are some formulations that do not feel right. I am unable to highlight those throughout the text, as I am not an expert on this, but just a small example in Table 1 "permanent and not permanent donor-recipient relationship" . I do not understand this term and I cannot imagine what it means. Or Line 336 "general interview scenario". I think you mean interview outline?

3. In the abstract the last few lines "The proposed research approach consisting in the use of mixed methodology in the fields of sociology and psychology in research on the phenomenon of living kidney donation is

definitely innovative" need to be changed. Even though the use of those methods are not very common in living organ donation, they cannot be presented anymore as "definitely innovative". I consider this an exaggeration (and I am doing such studies myself) and suggest to replace it with an other adjective.

I would really like to see this research published as I consider it important, the authors have done a commendable work, very time consuming and very thorough, and and I would be happy if they could address these rather easy to be addressed concerns.

My warmest regards

7. PLOS authors have the option to publish the peer review history of their article (what does this mean?). If published, this will include your full peer review and any attached files.

Reviewer #1: No

Reviewer #2: No

---

## [Author Response · Author response to Decision Letter 1]

22 Jan 2021

Thank you for the subsequent thorough and substantive evaluation of our article. We would also like to thank you for the words of praise expressed towards our work at the previous stage of the review procedure. This time, we have made every effort to respond to the submitted comments as honestly as possible.

We present below the authors' responses to the comments of both Reviewers. They concern both linguistic errors and inaccuracies as well as those in the substantive layer of the text. 

Response to comments from Reviewer No. 1

Abstract

1. Results: “lack of kidney in the body” would read better as “absence of a kidney in the body”

1. Thank you for this linguistic comment. We agree with this suggestion. The change has been applied.

2. Conclusions: final sentence is very wordy. May read better as: The proposed approach utilising mixed methodology within the fields of sociology and psychology for researching the phenomenon of living kidney donation is definitely innovative.

2. Thank you for proposing a better wording of this sentence. The change was introduced and combined with Reviewer No. 2's comment.

Main Text

3. Line 74 – “… negligible risk of physical and mental health disorders for living kidney donors”

4. Line 170 – The first concern that needs to be taken into account is the highly subjective…

3. and 4. Thank you for pointing out the lack of precision in the terminology used in lines 74 and 170. We have corrected the indicated phrases after re-examining the cited literature.

Table 1

5. Please clarify what is meant by “permanent / no permanent relationship”

5. Thank you for drawing attention to this vague sociodemographic category. It has been renamed and assigned clearer subcategories as well.

6. Line 213: should read “symphysis pubis”

7. Line 214: shorter pain is incorrect. Should read ‘pain is less’ or ‘pain is reduced’

8. Line 237: consent being obtained ‘verbally’ is better English

9. Line 320: should read “were more likely” (past tense)

6., 7., 8. and 9. We agree that the suggested language phrases and forms are more correct in the English language. All the corrections have been made. Thank you.

10. Line 333: Would read better as ‘None of the invited participants declined to participate in the interview, nor ended it prematurely’

10. Thank you for the better wording of this sentence. It has been applied.

Conclusions

11. Rather than use the word ‘goals’ I would refer back to the aims for the study – ‘Both of the aims of the study have been/were achieved’

11. We have made the suggested changes to this sentence, which makes it more precise. Thank you.

12. I’m unsure what ‘project realization’ is

12. We used the phrase 'carrying out the study'. We trust that in this way the meaning of this sentence is clearer. Thank you for making this comment.

Response to comments from Reviewer No. 2

1. The paper still feels too long and there is a lot of repetition mostly when it comes to the description of the method (especially grounded theory and the sociological background of the author). As much as I admire the passion of the first author regarding the methods, they way it is written makes it at least for me difficult to follow sometimes due to repetition, on expense of the results. Thus, I would recommend the authors identify repetitions and re-write in a more condensed manner.

1. The article was shortened in the 'Introduction' and 'Qualitative methods' parts. First of all, we shortened the threads concerning the description of the grounded theory methodology as well as the motives for its application in the study. We removed content redundancy. Thank you for this comment. We hope that the reading of the text will now be easier.

2. Even though I am not a native english speaker, I still think there are some formulations that do not feel right. I am unable to highlight those throughout the text, as I am not an expert on this, but just a small example in Table 1 "permanent and not permanent donor-recipient relationship" . I do not understand this term and I cannot imagine what it means. Or Line 336 "general interview scenario". I think you mean interview outline?

2. Thank you for drawing attention to this vague sociodemographic category. It has been renamed and assigned clearer subcategories as well.

As for line 336, in fact, we meant interview outline. Thank you for pointing out this linguistic inaccuracy. 

3. In the abstract the last few lines "The proposed research approach consisting in the use of mixed methodology in the fields of sociology and psychology in research on the phenomenon of living kidney donation is definitely innovative" need to be changed. Even though the use of those methods are not very common in living organ donation, they cannot be presented anymore as "definitely innovative". I consider this an exaggeration (and I am doing such studies myself) and suggest to replace it with another adjective.

3. We agree that the use of a mixed methodology in sociology and psychology in studying the phenomenon of living kidney donation is not innovative, it is simply not very common. Thank you for this remark.

---

## [Decision Letter · Decision Letter 2]

16 Feb 2021

PONE-D-20-15970R2

Experiencing one's own body and body image in living kidney donors – a sociological and psychological study

PLOS ONE

Dear Dr. Kowal,

Thank you for submitting your manuscript to PLOS ONE. I am happy to conditionally accept the paper, but it does not fully meet PLOS ONE’s publication criteria as it currently stands. Therefore, we invite you to submit a revised version of the manuscript that addresses the points raised during the review process.

ACADEMIC EDITOR:

Reviewers were generally happy with the revisions, but reviewer 1 still recommends a few grammatical changes / minor edits. In principle i'm happy to accept, if these changes are done satisfactorily.

We look forward to receiving your revised manuscript.

Kind regards,

Frank JMF Dor, M.D., Ph.D., FEBS, FRCS

Academic Editor

PLOS ONE

Reviewers' comments:

Reviewer's Responses to Questions

**Comments to the Author**

1. If the authors have adequately addressed your comments raised in a previous round of review and you feel that this manuscript is now acceptable for publication, you may indicate that here to bypass the “Comments to the Author” section, enter your conflict of interest statement in the “Confidential to Editor” section, and submit your "Accept" recommendation.

Reviewer #1: All comments have been addressed

Reviewer #2: All comments have been addressed

2. Is the manuscript technically sound, and do the data support the conclusions?

Reviewer #1: Yes

Reviewer #2: Yes

3. Has the statistical analysis been performed appropriately and rigorously? 

Reviewer #1: Yes

Reviewer #2: Yes

4. Have the authors made all data underlying the findings in their manuscript fully available?

Reviewer #1: Yes

Reviewer #2: Yes

5. Is the manuscript presented in an intelligible fashion and written in standard English?

Reviewer #1: Yes

Reviewer #2: Yes

6. Review Comments to the Author

Reviewer #1: Thank you for addressing the comment I have raised previously.

However, there remain a number of grammatical errors that require further amendment in order for the manuscript to be suitable for publication.

Line 74: The data from studies carried out by various transplant centers on a large group of donors indicate a low risk of complications during and after the donation procedure [1].

This should read: The data from studies carried out by various transplant centers on large groups of donors indicate a low risk of complications during and after the donation procedure [1].

Line 174: The first one concerns emphasizing the subjective nature of experiencing one's own body and the changes taking place in it [37].

This should read: The first one concerns emphasizing the subjective nature of experiencing one's own body and the changes taking place within it [37].

Table 1: The marital status still does not make sense and requires further clarification. You have categorised people as either married or single / and those not in permanent partnership relationships. I would either categorise these as 'married' and 'single' or you need to say 'married / in a long-term relationship' and 'single / not in a long-term relationship'. This will the provide a clear distinction between those two groups.

Reviewer #2: (No Response)

7. PLOS authors have the option to publish the peer review history of their article (what does this mean?). If published, this will include your full peer review and any attached files.

Reviewer #1: No

Reviewer #2: No

---

## [Author Response · Author response to Decision Letter 2]

16 Mar 2021

Thank you for the subsequent thorough and substantive evaluation of our article. This time, we have made every effort to respond to the submitted comments as honestly as possible.

1.

Line 74: The data from studies carried out by various transplant centers on a large group of donors indicate a low risk of complications during and after the donation procedure [1].

This should read: The data from studies carried out by various transplant centers on large groups of donors indicate a low risk of complications during and after the donation procedure [1].

Thank you for the better wording of this sentence. It has been applied.

2.

Line 174: The first one concerns emphasizing the subjective nature of experiencing one's own body and the changes taking place in it [37].

This should read: The first one concerns emphasizing the subjective nature of experiencing one's own body and the changes taking place within it [37].

Thank you for this linguistic comment. We agree with this suggestion. The change has been applied.

3.

Table 1: The marital status still does not make sense and requires further clarification. You have categorised people as either married or single / and those not in permanent partnership relationships. I would either categorise these as 'married' and 'single' or you need to say 'married / in a long-term relationship' and 'single / not in a long-term relationship'. This will the provide a clear distinction between those two groups.

Thank you for drawing attention to these vague sociodemographic categories. The changes have been applied.

---

## [Editor Report · Decision Letter 3]

18 Mar 2021

Experiencing one's own body and body image in living kidney donors – a sociological and psychological study

PONE-D-20-15970R3

Dear Dr. Kowal,

We’re pleased to inform you that your manuscript has been judged scientifically suitable for publication and will be formally accepted for publication once it meets all outstanding technical requirements.

Kind regards,

Frank JMF Dor, M.D., Ph.D., FEBS, FRCS

Academic Editor

PLOS ONE
---

## [Editor Report · Acceptance letter]

5 Apr 2021

PONE-D-20-15970R3 

Experiencing one's own body and body image in living kidney donors– a sociological and psychological study 

Dear Dr. Kowal:

I'm pleased to inform you that your manuscript has been deemed suitable for publication in PLOS ONE. Congratulations! Your manuscript is now with our production department. 

Kind regards, 

on behalf of

Dr. Frank JMF Dor 

Academic Editor

PLOS ONE